# Characterization of Genetic Diversity Conserved in the Gene Bank for Dutch Cattle Breeds

**Anouk E. van Breukelen** [1,*]**, Harmen P. Doekes** [1,2] **, Jack J. Windig** [1,2] **and Kor Oldenbroek** [1,2]

[1] Animal Breeding and Genomics, Wageningen University & Research, P.O. Box 338, 6700 AH Wageningen, The Netherlands; harmen.doekes@wur.nl (H.P.D.); jack.windig@wur.nl (J.J.W.); kor.oldenbroek@wur.nl (K.O.)

[2] Centre for Genetic Resources the Netherlands, Wageningen University & Research, P.O. Box 16, 6700 AA Wageningen, The Netherlands

**\*** Correspondence: anouk.vanbreukelen@wur.nl

**Abstract:** In this study, we characterized genetic diversity in the gene bank for Dutch native cattle breeds. A total of 715 bulls from seven native breeds and a sample of 165 Holstein Friesian bulls were included. Genotype data were used to calculate genetic similarities. Based on these similarities, most breeds were clearly differentiated, except for two breeds (Deep Red and Improved Red and White) that have recently been derived from the MRY breed, and for the Dutch Friesian and Dutch Friesian Red, which have frequently exchanged bulls. Optimal contribution selection (OCS) was used to construct core sets of bulls with a minimized similarity. The composition of the gene bank appeared to be partly optimized in the semen collection process, i.e., the mean similarity within breeds based on the current number of straws per bull was 0.32% to 1.49% lower than when each bull would have contributed equally. Mean similarity could be further reduced within core sets by 0.34% to 2.79% using OCS. Material not needed for the core sets can be made available for supporting in situ populations and for research. Our findings provide insight in genetic diversity in Dutch cattle breeds and help to prioritize material in gene banking.

**Keywords:** conservation; cryobank; genetic variation; cattle

## 1. Introduction

Genetic diversity refers to all genetic differences between species, between breeds within species and between individuals within breeds measured as differences in DNA [1]. Genetic diversity is essential for sustainable livestock production, because it provides the base material for genetic improvement of livestock and their adaptation to changing socio-economic and environmental demands. In addition to the management of genetic diversity in in situ populations, genetic diversity can be conserved ex situ in gene banks. Gene bank collections are important for three main reasons [1,2]: they (1) are an insurance against changes in market or environmental conditions; (2) are a safeguard against emerging diseases, political instability, and natural disasters; and (3) provide opportunities for research.

Prioritization of genetic material is an essential aspect of gene bank management, because financial and physical resources are generally limited [3]. Prioritization of material may occur at two levels. First, it has to be decided which animals from in situ populations are sampled to include in the gene bank. Second, genetic material within the gene bank can be divided into subsets based on their value for conservation. For the division of the total collection into subsets, different strategies may be used. One strategy is to set up a "core collection" for each breed, consisting of cryo-conserved samples that would allow the reconstitution of that breed, with an effective population size of at least 50 [4]. This strategy focuses on the conservation of genotypes in single breeds and on having a backup in

case of emergencies. It does not consider (overlap in) diversity across breeds. An alternative approach that focuses more on allelic diversity is to set up a "core set", following the method by Eding et al. [5]. Such a core set comprises a subset of the total collection that is optimized in terms of diversity. It could be defined as the smallest set of individuals that still encompasses the genetic diversity within that breed (or within multiple breeds combined).

Over the last century, genetic diversity in domesticated European cattle populations has been affected by two major factors. First, starting in the late 1960s, the Holsteinisation took place. This upgrading process resulted in the development of a Dutch Holstein Friesian population at the expense of native cattle breeds, as cows from native cattle breeds were inseminated by Holstein Friesian bulls (Figure 1) [6]. Small population sizes in native breeds increase the risk on inbreeding and loss of genetic diversity through genetic drift [7]. Second, artificial selection, particularly in the Holstein Friesian breed, facilitated by techniques like artificial insemination and embryo transfer in combination with cryo storage, has further reduced genetic diversity [8,9].

In 1975, three native dual-purpose breeds dominated the Dutch cattle population (Figure 1): Dutch Friesian cattle (76%), Meuse-Rhine-Yssel cattle (22%), and Groningen White Headed cattle (2%) [10]. Currently, more than 90% of the Dutch population consists of Holstein Friesian cattle [11]. After 1975, two additional native Dutch cattle breeds have been developed from the existing native MRY breed (Figure 1). These breeds were developed to conserve a breeding line with a characteristic color (Deep Red) and to develop a beef type breed (Improved Red and White). The herd books were established in 1988 and 2001, respectively.

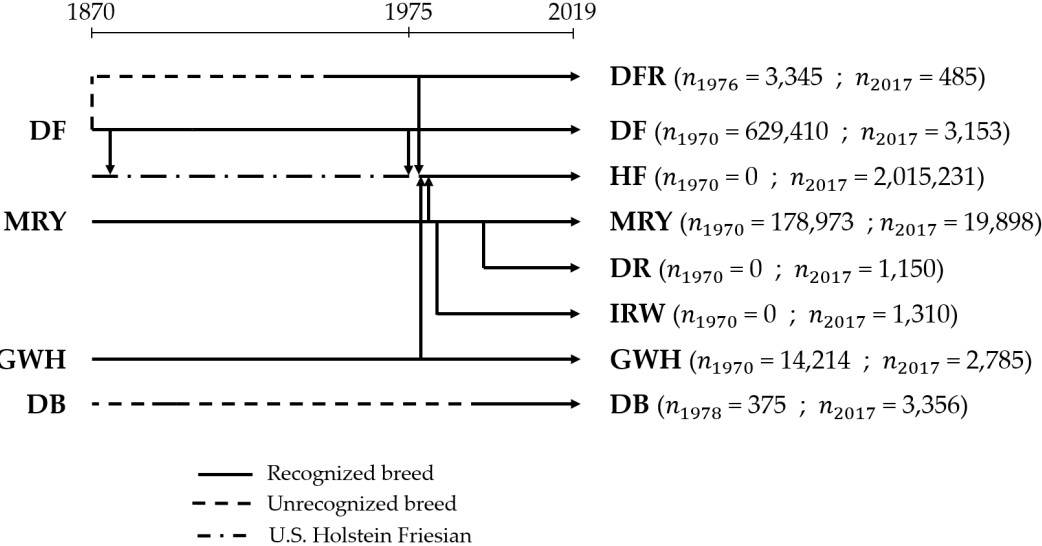

**Figure 1.** Development of Dutch cattle breeds from 1870 to 2019 and the number of cows per breed in 1970 (or in 1976 for DFR and 1978 for DB) and in 2017 (number of cows were based on [10,12,13]). DB: Dutch Belted; DF: Dutch Friesian; DFR: Dutch Friesian Red and White; DR: Deep Red; GWH: Groningen White Headed; IRW: Improved Red and White; MRY: Meuse-Rhine-Yssel.

Since the beginning of the 1990s, genetic material from the Holstein Friesian breed and from Dutch native cattle breeds has been collected and stored in the Dutch gene bank. For the Holstein Friesian breed, 25 straws have been stored for each AI bull. For native breeds, storage of material has been less systematic and has largely depended on the availability of samples (e.g., surpluses of semen from bulls used for AI before 1990) and financial resources. In later years, genetic material for native breeds has partly been collected to increase genetic diversity in the gene bank collection, based on (limited) pedigree information. Recently, all gene bank bulls from native breeds were genotyped. This enables a genomic analysis of the stored genetic diversity, a comparison across breeds and an evaluation of the composition of the current gene bank collection.

The objectives of this study were to (1) characterize genetic diversity conserved in the Dutch gene bank for native cattle breeds and the commercial Holstein Friesian breed, and (2) identify genetic material to set up core sets in which allelic diversity is optimized, either within or across breeds, and working sets with the remainder of material.

## 2. Materials and Methods

### 2.1. Data

The Dutch gene bank for livestock breeds is maintained by the Centre for Genetic Resources, the Netherlands (CGN) of Wageningen University and Research. From the CGN gene bank collections, a total of 715 bulls from seven Dutch native breeds, born between 1960 and 2015, were included in this study (after data filtering) (Figure 2). These bulls comprised all bulls from native breeds in the gene bank, except for a few bulls that were unsuccessfully genotyped. The seven breeds were Deep Red (DR), Dutch Belted (DB), Dutch Friesian (DF), Meuse-Rhine-Yssel (MRY), Friesian Red and White (DFR), Groningen White Headed (GWH), and Improved Red and White (IRW). Each bull was genotyped with the Illumina BovineSNP50 v2, Illumina BovineSNP50 v3 or Illumina BovineHD panel. After merging the different panels, 43,747 single nucleotide polymorphisms (SNPs) remained.

In addition to the native breeds, a sample of 165 Holstein Friesian (HF) gene bank bulls was included to be able to determine relatedness between HF and the other breeds. This sample consisted of available genotyped HF gene bank bulls born before 2000 ($n = 37$), as well as 25 black and 25 red HF gene bank bulls that were randomly sampled per five-year period after 2000 (Figure 2). The HF bulls were genotyped with the Illumina BovineSNP50 panel (versions v1 and v2) or the CRV custom-made 60 k Illumina panel (versions v1 and v2) and imputed to 75 k (for details, see Doekes et al. [9]). After merging the HF genotypes with those of the native breeds, which was done based on the SNP identifiers, 36,779 SNPs remained. Note that the HF sample did not cover all HF bulls in the gene bank. For a more extensive analysis of the HF gene bank collection, see Doekes et al. [9].

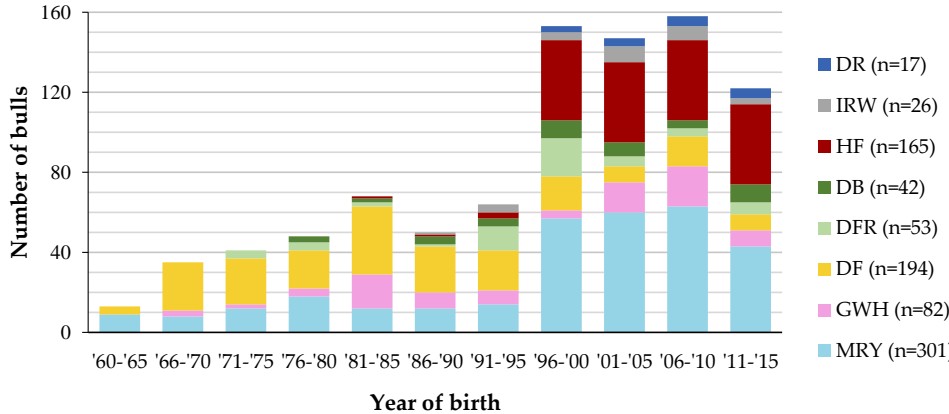

**Figure 2.** The number of bulls included in this study per breed and year of birth. DB: Dutch Belted; DF: Dutch Friesian; DFR: Dutch Friesian Red and White; DR: Deep Red; GWH: Groningen White Headed; IRW: Improved Red and White; MRY: Meuse-Rhine-Yssel.

The initial dataset, consisting of 880 bulls (735 bulls from native breeds and 165 HF bulls), was first filtered. Bulls with more than 10% missing SNPs ($n = 3$) were discarded. Bulls from native breeds with a HF fraction of 3/8 or more in the first three ancestral generations of their pedigree ($N_{DF} = 4$, $N_{DFR} = 4$, $N_{GWH} = 6$, $N_{MRY} = 3$) were also discarded, to prevent bulls with a high HF fraction being selected when constructing core sets for native breeds. In addition, SNPs with a call rate < 90% ($n = 171$) and SNPs that were non-polymorphic ($n = 1583$) were removed. After filtering, 35,025 SNPs remained. The density of SNPs was rather uniform across the genome (Supplement 1) with a mean physical distance between two successive SNPs of 69.08 Kb.

### 2.2. Characterization of Genetic Diversity within and between Breeds

Genetic diversity was quantified with genetic similarities. The similarity between any two bulls was calculated as [14]

$$SIM = \frac{\sum_{j=1}^{n_{SNP}}[I_{11} + I_{12} + I_{21} + I_{22}]}{4n_{SNP}},\tag{1}$$

where $n_{SNP}$ is the number of SNPs, and $I_{xy}$ is an indicator variable that, for the *jth* SNP, equaled 1 when allele $x$ of the first bull was identical to allele $y$ of the second bull, and 0 otherwise.

Note that the mean similarity (including self-similarity) equals expected homozygosity. Consequently, one minus the mean similarity equals the expected heterozygosity, which is commonly used as diversity measure [15]. A 1% higher mean similarity thus means a 1% lower expected heterozygosity at SNP level. Genetic distances between bulls were obtained as one minus the similarity between those bulls. These distances were used to construct a neighbor-joining (NJ) tree with the package Ape in R v3.3.3 [16]. The NJ-tree was visualized in FigTree [17].

Population structure was further investigated using the variational Bayesian framework implemented in fastSTRUCTURE [18]. Advantages of this approach are that population structure is not defined a priori (i.e., there are no predefined entities such as breeds) and that the number of clusters can be increased one by one, such that the development of clusters can be investigated and the uppermost likely number of clusters can be identified. A simple prior approach was used. The predefined number of clusters (K) ranged from two to ten. For each K, 50 independent runs were executed. From these runs, the uppermost likely K was identified based on the change in likelihood between runs for each K, following the approach of Evanno et al. [19]. Results were visualized in R with the POPHELPER package [20].

### 2.3. Optimizing Diversity in Collections

The mean similarity for a set of bulls was calculated following Berg and Windig [21] as

$$\overline{SIM} = c'Hc,\tag{2}$$

where $\overline{SIM}$ is the mean similarity, $H$ is the similarity matrix, and $c$ is a vector of proportional contributions that sum up to one.

We applied this formula to the entire collection of native breeds (all breeds combined) as well as to individual native breeds. To determine a core set of bulls we considered three different scenarios for the contributions: (1) equal contributions, (2) current contributions, and (3) optimal contributions. In scenario (1), each bull contributed equally. This scenario was used to determine the mean similarity if from each bull the same number of straws would be set aside for the core set. In scenario (2), each bull contributed based on the current storage of straws. This scenario was used to determine the mean similarity if from each bull straws would be set aside for the core set according to the contributions in the current collection. This scenario also provided information on whether the current composition of the gene bank was already optimized, i.e., whether breeds and/or bulls that are important for the diversity have contributed more to the gene bank. In scenario (3), each bull was given an optimal contribution based on optimal contribution selection (OCS), such that the core set was constructed to minimize the mean similarity. Although OCS was originally introduced to maximize genetic progress while restricting loss of diversity [22,23], it may also be used to optimize diversity irrespective of gain [9,24]. In OCS, each selection candidate is assigned an optimal contribution, which can be interpreted as the percentage of offspring that would minimize the mean similarity in the next (hypothetical) generation (in this study the contribution is used to determine the percentage of straws in the gene bank). In other words, it optimizes vector $c$ to minimize the similarity. Optimal contributions were calculated with GENCONT v2.0 [25].

## 3. Results

### 3.1. Characterization of Genetic Diversity within and between Breeds

The mean genetic similarity within breeds ranged from 64.24% (HF) to 69.05% (GWH) (Table 1). Between breeds, the mean similarity ranged from 61.82% (between DF and HF) to 65.51% (between DF and DFR). The IRW and DR breeds, which both descend from MRY (Figure 1), showed a high mean similarity with MRY: 63.67% and 64.76%, respectively. IRW and DR also showed a high similarity with each other (63.64%). HF bulls were least similar to all other breeds, with a mean similarity ranging from 61.81% to 61.91%.

The NJ-tree based on genetic distances (i.e., one minus genetic similarity) confirmed the abovementioned findings and provided additional insights for individual bulls (Figure 3). For example, various DFR bulls clustered within the DF cluster. Similarly, DR bulls clustered within MRY. The IRW formed a distinct cluster from MRY, despite its relatively high mean similarity with MRY. The GWH and HF were most distant from the other breeds.

**Table 1.** Mean genetic similarities (%) within breeds [1] (on diagonal, including self-similarities) and between breeds (below diagonal).

| Breed | DB | DF | DFR | DR | GWH | HF | IRW | MRY |
|-------|-------|-------|-------|-------|-------|-------|-------|-------|
| DB | 66.87 | | | | | | | |
| DF | 63.61 | 66.71 | | | | | | |
| DFR | 63.58 | 65.51 | 66.47 | | | | | |
| DR | 63.04 | 62.89 | 63.01 | 65.80 | | | | |
| GWH | 62.89 | 62.97 | 63.02 | 62.97 | 69.05 | | | |
| HF | 61.91 | 61.82 | 61.89 | 61.83 | 61.91 | 64.24 | | |
| IRW | 63.04 | 63.06 | 63.11 | 63.64 | 63.02 | 61.75 | 64.69 | |
| MRY | 63.11 | 62.78 | 62.92 | 64.76 | 62.93 | 62.02 | 63.67 | 66.44 |

[1] DB: Dutch Belted; DF: Dutch Friesian; DFR: Dutch Friesian Red and White; DR: Deep Red; GWH: Groningen White Headed; HF: Holstein Friesian; IRW: Improved Red and White; MRY: Meuse-Rhine-Yssel.

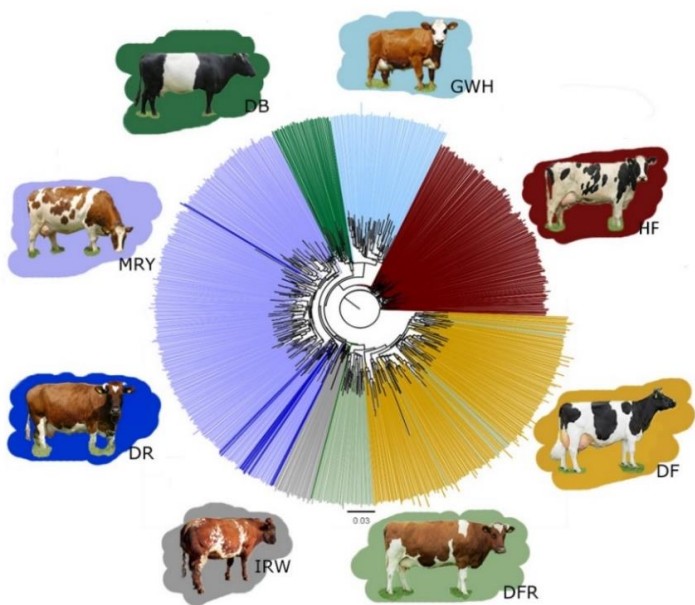

**Figure 3.** Neighbor-joining tree based on genetic distances (one minus genetic similarity) between individual bulls (each line is a bull). Each breed is shown with a distinct color. DB: Dutch Belted; DF: Dutch Friesian; DFR: Dutch Friesian Red and White; DR: Deep Red; GWH: Groningen White Headed; HF: Holstein Friesian; IRW: Improved Red and White; MRY: Meuse-Rhine-Yssel.

The fastSTRUCTURE results provided additional information on population admixture and breed divergence (Figure 4). The most likely number of clusters was K = 4, based on the change in likelihood (Supplement 2). At K = 4, each of the three native breeds that were originally the most common breeds in the Netherlands (i.e., DF, GWH, and MRY) was part of a different cluster, and HF formed the fourth cluster. When adding additional clusters, at K = 5, DB bulls were assigned to the fifth cluster. At K = 8, however, the three remaining breeds (DR, MRY, and DFR) did not form clusters of their own. Instead, at K = 6 the cluster containing both DF and DFR bulls was split into two clusters, but these clusters did not separate the two breeds. At K = 7, a 3rd cluster was formed in the DF – DFR cluster, and this cluster mainly contained DFR bulls, but also some DF bulls. At K = 8, MRY, IRW, and DR bulls were partly assigned to the eighth cluster, but still IRW and DR bulls did not cluster separately and shared most variation with the MRY bulls. In fact all DR and IRW bulls were a mixture of 4–5 different clusters.

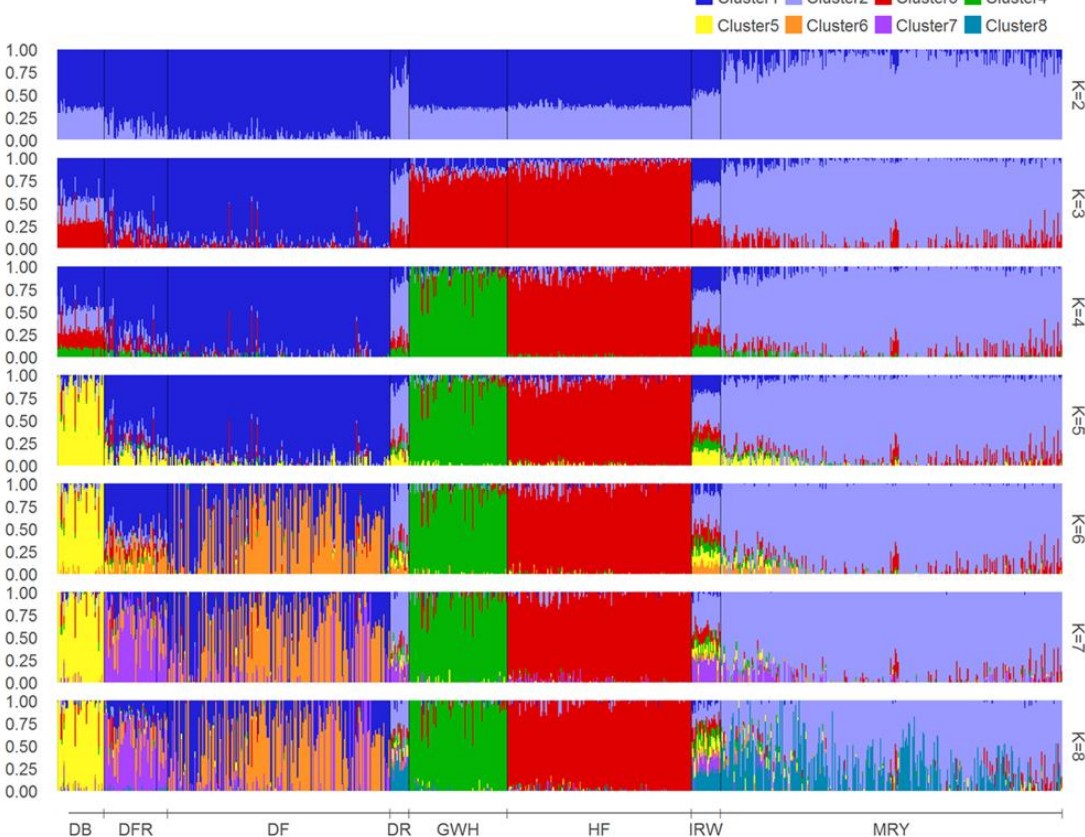

**Figure 4.** Population structure as estimated by fastSTRUCTURE divided into 2–8 clusters (K) for each breed (ordered alphabetically). DB: Dutch Belted; DF: Dutch Friesian; DFR: Dutch Friesian Red and White; DR: Deep Red; GWH: Groningen White Headed; HF: Holstein Friesian; IRW: Improved Red and White; MRY: Meuse-Rhine-Yssel.

### 3.2. Optimization of Diversity across Breeds

The mean similarity of all bulls across the breeds based on equal contributions was 68.06% while similarity based on current contributions was 66.38%. When performing OCS, the mean genetic similarity further decreased to 64.78%. From the in total 715 bulls, 72 bulls received an optimal contribution higher than zero. These 72 bulls would be prioritized when the aim is to set up a core set collection that is optimized to conserve allelic diversity across native breeds.

The relative contributions of breeds differed across the three scenarios. For example, when considering equal contributions across bulls, the summed contribution was highest for MRY, namely, 42% (Figure 5). This is because MRY simply had the largest number of bulls used in this study (Figure 2).

When considering the current storage of straws per bull, however, the summed contribution of MRY (25%) was lower than that of DF (of 28%). This is because the average number of straws per MRY bull was relatively low compared to the other breeds. When performing OCS, the summed contribution of MRY was even lower, namely, 16%. This suggests that, when the aim is to set up a core set to conserve allelic diversity across native breeds, the relative contribution of MRY should be lower than the current contributions in the entire collection, whereas the relative breed contributions should be increased for DB (from 2% to 10%), DF (28% to 37%), and IRW (2% to 17%), and decreased for DFR (19% to 3%), DR (6% to 4%), and GWH (18% to 13%).

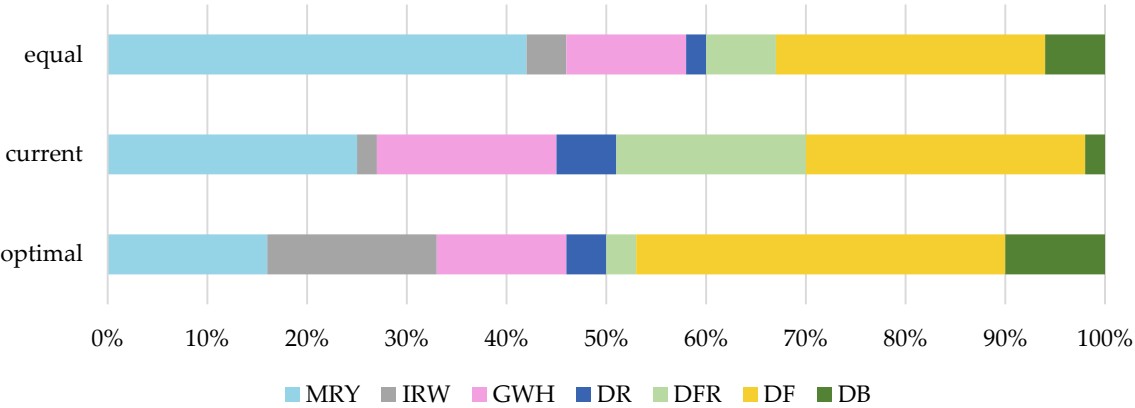

**Figure 5.** Summed contribution per native breed in the gene bank based on equal contributions across bulls, current contributions based on straws per bull, and optimal contributions based on minimizing the mean genetic similarity. DB: Dutch Belted; DF: Dutch Friesian; DFR: Dutch Friesian Red and White; DR: Deep Red; GWH: Groningen White Headed; IRW: Improved Red and White; MRY: Meuse-Rhine-Yssel.

*3.3. Optimization of Diversity within Breeds*

Within each breed, the mean similarity based on current contributions was lower (ranging from 0.32% for MRY to 1.49% for DR) than the mean similarity for equal contributions (Table 2). When contributions were optimized, e.g., to set up a core set that is optimized to conserve allelic diversity within the breed, the mean similarity could be further reduced, ranging from 0.34% for DR, to 2.79% for IRW.

When optimizing contributions within native breeds, bulls from all ages were selected, with some differences across breeds (Table 3. For the majority of breeds, the summed contribution for bulls born before 2000 was higher than those for bulls born after 2000. More specifically, the summed contribution for bulls born before 2000 was 64% in DB, 68% in DF, 74% in DFR, 59% in IRW, and 57% in MRY. For DR, all bulls were born after 2000. For GWH, most of the contributions (64%) were assigned to bulls born after 2000. These findings suggest that old bulls harbor valuable genetic diversity not present in more recent bulls, with some differences across breeds.

**Table 2.** Mean similarity (%) within breeds [1] based on equal contributions, current contributions and optimal contributions. The difference between the mean similarity for current and equal contributions, and for optimal and current contributions is also shown.

| Breed | Equal | Current | Optimal | Current-Equal | Optimal-Current |
|-------|-------|---------|---------|---------------|-----------------|
| DB | 66.87 | 65.57 | 64.33 | −1.3 | −1.24 |
| DF | 66.71 | 65.92 | 63.36 | −0.79 | −2.56 |
| DFR | 66.47 | 65.24 | 64.77 | −1.23 | −0.47 |
| DR | 65.80 | 64.31 | 63.97 | −1.49 | −0.34 |
| GWH | 69.05 | 67.91 | 66.23 | −1.14 | −1.68 |
| IRW | 64.69 | 64.36 | 61.57 | −0.33 | −2.79 |
| MRY | 66.44 | 66.12 | 64.56 | −0.32 | −1.56 |

[1] DB: Dutch Belted; DF: Dutch Friesian; DFR: Dutch Friesian Red and White; DR: Deep Red; GWH: Groningen White Headed; IRW: Improved Red and White: MRY: Meuse-Rhine-Yssel.

**Table 3.** The number of selection candidates (n), the number of selected bulls (sel), and the summed contribution for selected bulls (c, in %) per birth period when performing optimal contribution selection within native breeds [1].

| Birth Year | DB | | | DF | | | DFR | | | DR | | | GWH | | | IRW | | | MRY | | |
|------------|----|-----|----|----|-----|----|-----|-----|----|----|-----|----|-----|-----|----|----|-----|----|-----|-----|----|
| | n | sel | c | n | sel | c | n | sel | c | n | sel | c | n | sel | c | n | sel | c | n | sel | c |
| 1960–1969 | | | | 24 | 4 | 11.9 | | | | | | | 2 | 1 | 0.6 | | | | 16 | 5 | 13.9 |
| 1970–1979 | 3 | 1 | 0.8 | 44 | 7 | 10.8 | 6 | 4 | 9.9 | | | | 5 | 1 | 2.6 | | | | 29 | 14 | 28.4 |
| 1980–1989 | 6 | 6 | 36.9 | 47 | 9 | 29.5 | 2 | 1 | 0.6 | | | | 23 | 6 | 22.8 | 1 | 1 | 1.1 | 20 | 6 | 3.5 |
| 1990–1999 | 10 | 9 | 26.1 | 45 | 4 | 16.1 | 28 | 22 | 63.6 | | | | 10 | 2 | 9.6 | 6 | 5 | 57.8 | 64 | 7 | 11.2 |
| 2000–2009 | 14 | 11 | 16.2 | 20 | 4 | 31.6 | 8 | 7 | 11.1 | 11 | 11 | 62.7 | 32 | 14 | 41.6 | 13 | 13 | 27.0 | 123 | 14 | 22.5 |
| 2010–2015 | 9 | 6 | 20.0 | 10 | 0 | 0 | 9 | 8 | 14.8 | 6 | 6 | 37.3 | 10 | 7 | 22.7 | 6 | 6 | 14.0 | 51 | 14 | 20.5 |

[1] DB: Dutch Belted; DF: Dutch Friesian; DFR: Dutch Friesian Red and White; GWH: Groningen White Headed: MRY: Meuse-Rhine-Yssel.

## 4. Discussion

### 4.1. Characterization of Cattle Breeds in the Dutch Gene Bank

We observed that HF bulls were on average the least similar to bulls from native Dutch breeds (Table 1). This suggests that the native breeds harbor genetic variation that is not present in HF, and the other way around. In addition to HF, the three historically large native breeds, DF, MRY, and GWH, formed the most distinct clusters (Figures 3 and 4). The other breeds either clustered within one of the older breeds, or showed clear admixture.

As expected from their breed development (Figure 1), IRW and DR bulls had a high mean similarity with MRY bulls (Table 1). The IRW bulls, however, clustered separately in the NJ tree (Figure 3). The distinctiveness of IRW may be explained by some influence of Belgian Blue (BBL) cattle, which have been used to improve meat quality traits in the IRW. Indeed, BBL ancestors were identified in the pedigree of a few of the IRW bulls, although the fraction of BBL was rather small with a maximum of 1/8 for the last three ancestral generations. The DR clustered only partially together with MRY, which could be explained by DR being developed more recently compared to the IRW (Figure 1) and DR being a dual-purpose breed like MRY.

We observed that DB had a relatively high mean similarity with DF and DFR bulls (Table 1), which was also observed by Eding et al. [5]. As registration for DB cattle started only in 1997, it is likely that before 1997 part of the DB cattle were upgraded by breeding DF females with DB bulls. The distinctiveness of DB is expected to be caused by random drift and the influence of American DB cattle, as various American DB ancestors were identified in the pedigree of DB gene bank bulls. The American DB is known to be partially founded by Dutch DB cattle, which were exported from the Netherlands in 1838, 1840, and 1858 [12].

Based on the multiple clusters that seemed to be present in DF cattle and MRY cattle in the fastSTRUCTURE results at K-levels above 6 (Figure 4), we decided to perform a principal component analysis (PCA) for DF and MRY. The PCA was based on a genomic relationship matrix in PLINK v1.9 [26]. For DF, a distinct group of eighteen bulls was identified on the second principal component (Supplement 3.1). Eleven out of the eighteen bulls in this cluster were from one of the so-called "fundament breeders". Since 1992, the DF breed society has applied fundament breeding, in which fundament breeders use their own bulls for breeding as much as possible [27]. The objective of this approach is to maintain genetic diversity by creating different groups of breeding animals, with each their own unique genetic diversity and a low kinship between groups. Ten breeders have been recognized as DF fundament breeder; three of which had more than one bull in the gene bank and five of which had a single bull in the gene bank. Based on the first two principal components, at least two out of three large fundament breeding groups appeared to offer unique genetic diversity within the DF breed (Supplement 3.1). This finding gives incentive to make efforts to collect material from such fundament breeders for storage in the gene bank. For MRY, no visual subclusters were observed on the first two principal components (Supplement 3.2).

### 4.2. Management and Optimization of Genetic Diversity within Breeds

The management and optimization of genetic diversity within breeds is relevant when conservation efforts are directed towards the conservation of individual breeds, to conserve their unique combination of alleles or genotypes. The conservation of breeds in gene banks facilitates the restoration of a breed in case of a disease outbreak or accumulation of recessive disorders due to inbreeding [28]. Within breeds, we first showed that the mean genetic similarity based on the current storage of straws was lower (0.32% to 1.49%, depending on the breed) than the mean similarity when each bull contributed equally (Table 2). The current composition of the gene bank appeared partly optimized in terms of genetic diversity based on pedigree information. A further reduction in the mean genetic similarity (of 0.34% to 2.79%, depending on the breed) could be achieved when using optimal contributions. Bulls with an optimal contribution larger than zero can be included in a core set for the specific breed. The number

of straws stored in the core set across bulls should be in the same ratio as the optimal contributions to minimize the mean similarity in the core set. Note that this is sometimes not possible due to practical limitations (e.g., when a bull with a high optimal contribution has only few straws available). Material from bulls with an optimal contribution of zero can be relocated to a working set, where the material is accessible for, among others, the support of in situ populations, research, introgression of specific traits into breeds, and development of new breeds [28]. We furthermore recommend using OCS to determine which bulls from the in situ populations should be included to the gene bank. This requires the analysis of genomic information (or well documented pedigree information) of both the ex situ and in situ populations.

### 4.3. Value of Old and Recent Germplasm Material

We observed that both "old" bulls (i.e., bulls born before 2000) and recent bulls were selected when optimizing genetic diversity (Table 3). This suggests that also old gene bank bulls harbor valuable genetic diversity for in situ populations, although genomic information from in situ populations is required for further validation. A drawback of using old bulls is that, as a result of selection, these bulls are expected to have lower breeding values than bulls that were born more recently. Thus, by introducing their material, old bulls may increase genetic diversity at the cost of genetic merit. Previous studies have shown, however, that old gene bank bulls can be effectively used to maximize genetic merit for a given level of diversity [24,29]. For the MRY breed, Eynard et al. [24] showed that the Dutch-Flemish total merit index (NVI) could be increased by a few points when using old gene bank bulls (born before 2000) in addition to current bulls (born after 2000). For the HF breed, Doekes et al. [29] found that the benefit of using gene bank bulls in addition to current AI-bulls depends on (1) the relative emphasis on genetic diversity and (2) the selection criterion. As expected, the relative benefit of using gene bank bulls was found to be larger when more emphasis was put on genetic diversity. Furthermore, the benefit was relatively small when selecting for the total merit index NVI, but higher when selecting for a specific index, such as fertility. Doekes et al. [29] concluded that, anticipating changes in breeding goal in the future (as result of changing environments and changing market demands), the gene bank collection is a valuable resource in terms of both genetic diversity and genetic merit.

### 4.4. Management of Genetic Diversity across Breeds

The management and optimization of genetic diversity across breeds is relevant when conservation efforts are directed towards conservation of overall allelic diversity and not necessarily conservation of the different combinations of alleles (and thereby phenotypes). Across breeds, there seemed to be substantial overlap in the conserved genetic diversity. For example, when optimizing genetic diversity across all native breeds combined, only 72 out of 715 bulls received an optimal contribution above zero. This overlap in genetic diversity across breeds is not surprising, when considering breed history (Figure 1). To investigate the influence of HF on the optimal contributions of the native breeds, we also performed OCS including HF bulls. Inclusion of HF bulls lowered the contributions of native breeds, because part of the contributions (37%) was assigned to HF bulls. The relative contributions of native breeds, however, remained very similar.

DFR bulls had a high genetic similarity to DF bulls (Table 1) and the optimal contribution of DFR when maximizing genetic diversity across breeds was only 1% (Figure 5). Based on these findings, one may question whether efforts should be made to conserve breeds like DF and DFR separately or whether they should be managed as a single breed. By managing them as a single breed, the population size increases, which could help to decrease inbreeding and drift effects.

In this study, we only considered the optimization of a single gene bank in a single country. However, there may also be overlap in the genetic diversity that is stored in gene banks worldwide. For the globally connected HF breed, for example, Danchin-Burge et al. [8] showed substantial overlap between US, French, and Dutch germplasm collections. However, they also indicated that there are

various arguments in favor of this "redundancy". From a safety perspective it might be wise to have duplo-collections as are common in plant genetic resources. From a policy perspective, each country is supposed to manage its own genetic resources. From a practical point of view, germplasm stored in a national gene bank is more readily available, as exchange of animal germplasm over national borders must comply with international regulations, such as veterinary regulations and access and benefit sharing regulations under the Nagoya Protocol. However, gene bank collections are costly. Recently, de Oliveira Silva et al. [30] developed a mathematical model to optimize logistical decisions of conserving breeds in terms of economics. They evaluated alternative scenarios for reallocating genetic material currently stored in different European gene banks and showed that overall costs may be reduced by ~20% by selecting gene banks that have a relatively low combination of fixed and collection costs. Further work in this area would be valuable to economically and genetically optimize national and international gene bank collections.

Although the overlap in genetic diversity is expected to be less pronounced for native breeds compared to a transboundary breed like HF, there may still be some double-storage of genetic material across gene banks and/or countries. A major reason for this overlap might be the introgression of (e.g., transboundary) breeds into other (e.g., local) breeds. In this study, we partly corrected for the influence of HF by excluding bulls from native breeds with a HF fraction of 3/8 or more in the first three ancestral generations of their pedigree. Other breeds, however, may also have had an impact. For example, IRW bulls were found to have a fraction of BBL in their pedigree, which was likely part of the reason that IRW was assigned a relatively high contribution (17%) when performing OCS in Dutch native breeds (Figure 5). The fraction BBL that may be unique in the Dutch gene bank is likely to be well covered in the BBL population in Belgium. To account for influences of breeds like HF and BBL in native cattle populations, OCS can be extended to minimize migrant contributions while maximizing genetic diversity [31]. In this extended OCS approach, a reference population (consisting of breeds that are likely to have contributed to the native breeds) is used to determine the migrant contributions. In future work, it would be valuable to consider these migrant contributions when optimizing gene bank collections.

### *4.5. From Genotype to Sequence Data*

As this study is based on 35 k SNP data, it is likely that rare variants are not considered (see, e.g., in [32]). The effect of missing genetic variation may be stronger for local breeds than for a mainstream breed like HF, because of ascertainment bias (see, e.g., in [33]). Sequencing costs are continuously decreasing and, as a result, increasing sequencing data will be available. This data will be extremely valuable for gene bank collections, since it will help, for example, to identify rare genetic variants that were lost over time or variants that are unique to specific breeds. The developed procedures in this study will be applicable to optimize gene banks based on sequence data.

### 5. Conclusions

Based on genotype data, bulls from native Dutch cattle breeds stored in the national gene bank are genetically distinct from the random sample of HF gene bank bulls. Old bulls (born before 2000) contribute considerably to the genetic diversity in the gene bank. Within breeds, the current collection is already partly optimized to maximize allelic diversity. Core set collections could be set up using OCS based on genomic information. Across breeds, there is substantial overlap in the genetic diversity that is conserved in the gene bank. The increasing availability of genomic information and recent developments on economic modeling of gene bank collections and extension of OCS methodology may help to further optimize gene bank collections.

**Supplementary Materials:** The following are available online at http://www.mdpi.com/1424-2818/11/12/229/s1. Supplement 1: SNP density for each Bos Taurus Autosome (BTA); Supplement 2: Estimation of uppermost number of clusters according to Evanno et al. [19]; Supplement 3: (1) Principal component analysis within Dutch Friesian (DF) bulls showing groups of bulls from fundament breeders, and (2) Principal component analysis within MRY bulls.

**Author Contributions:** Conceptualization, A.E.V.B., H.P.D., J.J.W., and K.O.; Methodology, A.E.V.B., H.P.D., and K.O.; Formal Analysis, A.E.V.B.; Writing—Original Draft Preparation, A.E.V.B. and H.P.D.; Writing—Review & Editing, A.E.V.B., H.P.D., J.J.W., and K.O.; Visualization, A.E.V.B. and H.P.D.

**Funding:** The research leading to these results has been conducted as part of the IMAGE project, which received funding from the European Union's Horizon 2020 Research and Innovation Program under the grant agreement n° 677353. The study was co-funded by the Dutch Ministry of Agriculture, Nature and Food Quality (KB-34-013-002).

**Acknowledgments:** The authors gratefully acknowledge the Dutch-Flemish cattle improvement cooperative (CRV) for providing genotype data of Holstein Friesian bulls.

**Conflicts of Interest:** The authors declare no conflict of interest.

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
