# Peer review of "Characterization of Genetic Diversity Conserved in the Gene Bank for Dutch Cattle Breeds"

_diversity, doi:10.3390/d11120229_

Round 1

Reviewer 1 Report

Interesting paper - reads well. 

Some repetition of results and discussion, but not negative to the overall reading experience.

One minor remark - in abstract the reference to genotypes based on 35k is confusing as several arrays were used for genotyping and the 35k was the markers remaining after QC. Suggest to remove in abstract as it is clearly explained in the material & methods.

Author Response

One minor remark - in abstract the reference to genotypes based on 35k is confusing as several arrays were used for genotyping and the 35k was the markers remaining after QC. Suggest to remove in abstract as it is clearly explained in the material & methods.

Thank you for your useful comment. We have removed the number of SNPs in the abstract.

Reviewer 2 Report

Van Beukelen et al / Diversity (manuscript)

The paper is on a study on Dutch local cattle breeds (and their derivatives) and main stream dairy breed Holstein. The extant bull collection in a cryo storage is used to assess the current state of variation with respect to maximizing the variation in the storage itself and in the outcome from the use of the stored bulls. Exact measures on the variation are given.

General comments:

The study is neat (and harmless) with sufficient body of high quality data. Rather mild, observational and much used conclusions appear at the end.

The authors prefer the word diversity to variation – it could be interesting to hear why and add related comments on this in the introduction. Genomic similarity figures, e.g. in Table 1 and 2, range from 62 to 67%. Are these close to expected levels? These look rather similar. Are they different from each other with some statistical probability? What means a difference of 1 %-unit (cf Table 2) in this context – serious or meaningless difference? Please, advice the reader. It is very nice to see that the Holstein bulls are included in the study to provide a common ground for further comparisons. However, it is not clear why only bulls born before 2000 are included where Holstein is part of the analyses. The results on the current generation would have been interesting to see. The Holsteins’ genetic similarity is 64% (Table 1) - is this satisfactory? Towards the end, Holsteins disappear when it comes to analyzing the optimum amount of variation now or in the future (sections 3.2 and 3.3). This was a frustrating area in the study. I would encourage extending this part also to Holsteins. You mention the use of pedigree data in earlier decisions to build cryo storage. Do these breeds have useful pedigree data? Recently there have been papers on the use of genomic vs pedigree data in quantifying and maintaining diversity, e.g. very recent ones GSE 51, 31 and even JABG on-line published Toro et al. You probably have data to make a similar comparison. That would make the study much more interesting.

Minor issues:

There are two new breeds DR and IRW created from MRY. Do they share pedigree to quantify the differences on this ground as well? Figure 5 – their sum is rather stable across the study angles.

Of the factors affecting diversity of European cattle populations (lines 45-58), I would also mention the pressure to increase cows’ production performance and use available new technology (e.g. AI and ET)

Figure 1 is illuminating the breed development well. It gives the cow numbers in 2017. There is room to have them also at some earlier time point – in 1960 or alike. There are several arrows ending at Holstein Friesian in 1975 – the story would benefit from explaining them in the text.

Line 80: 715 bulls, line 100: 735 bulls – why are these figures different.

Formula 112 looks elaborated although it is simply 2 ZZ’/n with p’s equaling 0.5. Further, if this result is incorporated in 119, what do we have then for SIM.

Figure 3 has good looking cows and different colours for trees. On the whole, it has very little information for a scientific article. What are the dark blue branches telling?

Figure 4. Your story says that DR should be next to IRW (next to MRY) in this illustration.

Section 4.3. The optimization could include also breeding values – add a comment on this.

Lines 361-362. What extend the differences among local breeds are affected by differentiated introgression of external breeds into them. The study could be better if there were also other than Holstein included to arrive at a more comprehensive picture. E.g. Belgian Blue is mentioned in this sense (262).

Line 187: DRF -> DFR; Line 359: expected … expected (reduce redundancy)

Author Response

The authors prefer the word diversity to variation – it could be interesting to hear why and add related comments on this in the introduction.

Thank you for your useful comments. We use “genetic diversity”, because it is a commonly used term in the livestock sector, as well as in this journal (see e.g. a previous special issue on “Use of Molecular Markers in Genetic Diversity Research”). We now define genetic diversity in the first lines of the introduction.

Genomic similarity figures, e.g. in Table 1 and 2, range from 62 to 67%. Are these close to expected levels? These look rather similar. Are they different from each other with some statistical probability? What means a difference of 1 %-unit (cf Table 2) in this context – serious or meaningless difference? Please, advice the reader.

The mean similarity in a population is equal to one minus expected heterozygosity (2pq). Thus, a 1% increase in similarity means a 1% decrease in expected heterozygosity at the SNPs. Expected heterozygosity (Nei, 1973) is commonly used as a measure of genetic diversity. The values we observed are close to expected levels and to values observed previously (e.g. Engelsma et al. 2012 in JABG; Doekes et al. 2018 in JDS). They are also in line with what may be expected based on allele-frequency-distributions of the SNPs. Since allele-frequency-distributions of SNP data in cattle are approximately uniform (see e.g. Eynard et al. 2015 in BMC Genetics), the expected heterozygosity will be approximately equal to 1/3 and the mean similarity to 2/3, so 66.67%. We now describe the meaning of an increase of 1% in similarity and refer to the paper of Nei (1973) in the material and methods.

It is very nice to see that the Holstein bulls are included in the study to provide a common ground for further comparisons. However, it is not clear why only bulls born before 2000 are included where Holstein is part of the analyses. The results on the current generation would have been interesting to see.

For the years following 2000, a random sample of 25 red Holstein Friesian bulls and 25 black Holstein Friesian bulls were added for every five year period (as described in the material and methods).

The Holsteins’ genetic similarity is 64% (Table 1) - is this satisfactory?

It is not clear to us what is meant by “satisfactory” (?). The value was not surprising, since we found similar values in previous work (Doekes et al. 2018 in JDS).

Towards the end, Holsteins disappear when it comes to analyzing the optimum amount of variation now or in the future (sections 3.2 and 3.3). This was a frustrating area in the study. I would encourage extending this part also to Holsteins.

As suggested, we ran the same analyses including the sample of Holstein Friesian bulls. We now describe the findings in the Discussion. We decided not to include the analysis in the results because the HF breed is a large commercial breed which is stored in the gene bank for different purposes (e.g. a backup in case interesting bulls are untimely lost) than the native rare breeds.

You mention the use of pedigree data in earlier decisions to build cryo storage. Do these breeds have useful pedigree data? Recently there have been papers on the use of genomic vs pedigree data in quantifying and maintaining diversity, e.g. very recent ones GSE 51, 31 and even JABG on-line published Toro et al. You probably have data to make a similar comparison. That would make the study much more interesting.

We agree it would be interesting to compare pedigree and genomic data. However, the pedigree data of native breeds is not complete and not directly available. Especially for old bulls, pedigree data often does not go further than one generation back. In addition, pedigree data does not enable a comparison between breeds, which is why the comparison based on genotype data is valuable.

Previously, a comparison has been made between priorization based on optimal contributions with pedigree data and genomic data (Engelsma, et al., 2011). In this study it was shown that selection with genomic kinships resulted in a slightly higher conserved diversity.

Minor issues:

There are two new breeds DR and IRW created from MRY. Do they share pedigree to quantify the differences on this ground as well? Figure 5 – their sum is rather stable across the study angles.

It is known that DR and IRW bulls have MRY ancestors. However, because the pedigree data of native breeds is not complete and not directly available we did not analyze pedigree data.

Of the factors affecting diversity of European cattle populations (lines 45-58), I would also mention the pressure to increase cows’ production performance and use available new technology (e.g. AI and ET)

A line has been added to include AI and ET.

Figure 1 is illuminating the breed development well. It gives the cow numbers in 2017. There is room to have them also at some earlier time point – in 1960 or alike.

We now show cow numbers at an earlier point in Figure 1.

There are several arrows ending at Holstein Friesian in 1975 – the story would benefit from explaining them in the text.

We now describe the influence of native breeds in the creation of the Dutch Holstein Friesian population.

Line 80: 715 bulls, line 100: 735 bulls – why are these figures different.

The number 715 was after data filtering, for which the steps are explained in a later paragraph. It is now described between brackets that the number 715 is after data filtering.

Formula 112 looks elaborated although it is simply 2 ZZ’/n with p’s equaling 0.5. Further, if this result is incorporated in 119, what do we have then for SIM.

The formulas have been replaced by a single formula that is simpler, especially for readers that are less familiar with the genomic relationship context.

Figure 3 has good looking cows and different colours for trees. On the whole, it has very little information for a scientific article. What are the dark blue branches telling?

We believe the figure is informative, since it shows the genetic distances between individual gene bank bulls (in addition to the mean similarities reported in Table 1). For example, the dark blue branches show that some DR bulls are more similar to MRY bulls than to other DR bulls.

Figure 4. Your story says that DR should be next to IRW (next to MRY) in this illustration.

In this figure the breeds are ordered alphabetically, which we now describe in the figure description. The colors of the clusters show which breeds are similar (MRY, DR and IRW share, among others, clusters 2 and 8).

Section 4.3. The optimization could include also breeding values – add a comment on this.

In the material and methods we address that optimal contribution selection is usually used to maximize the mean breeding value. In the discussion (4.3) we address that old gene bank bulls may have lower breeding values than recently born bulls.

Lines 361-362. What extend the differences among local breeds are affected by differentiated introgression of external breeds into them. The study could be better if there were also other than Holstein included to arrive at a more comprehensive picture. E.g. Belgian Blue is mentioned in this sense (262).

We agree that it would be interesting to look at introgression of breeds other than Holstein. However, we did not have genotype data available for other breeds.

Line 187: DRF -> DFR; Line 359: expected … expected (reduce redundancy)

The typo has been corrected and redundancy has been removed.

Round 2

Reviewer 2 Report

I am impressed about the prompt and well covered response by the authors.

When we quantify the genetic variation, often the figures are not very informative.  Measures like level of inbreeding, and although for somewhat different reasons in this case, level of heterozygosity. As the authors know, Ne is a useful in this context. Now the found levels of heterozygosity are over a wide range of values.  Do individual figures reflect history of admixture, bottlenecks in population size, very intensive selection etc.  That's why I was curious to understand the expected levels or reasons for deviations among the populations.

The authors use symbol H for similarity matrix which is probably made of SIM values produced earlier in the paper with the new clear formula.  That connection - if so - is missing.